# Risk factors for early PICC removal: A retrospective study of adult inpatients at an academic medical center

**Burton H. Shen**[1,2], **Lindsey Mahoney**[1,2], **Janine Molino**[3], **Leonard A. Mermel**[1,2,4] *

**1** Department of Medicine, Warren Alpert Medical School of Brown University, Providence, RI, United States of America, **2** Department of Medicine, Rhode Island Hospital, Providence, RI, United States of America, **3** Lifespan Biostatistics Core, Lifespan Hospital System, Providence, RI, United States of America, **4** Division of Infectious Diseases, Rhode Island Hospital, Providence, RI, United States of America

* lmermel@lifespan.org

## Abstract

### Background

Use of PICCs has been rising since 2001. They are used when long-term intravenous access is needed and for blood draws in patients with difficult venous access.

### Objective

To determine which risk factors contribute to inappropriate PICC line insertion defined as removal of a PICC within five days of insertion for reasons other than a PICC complication.

### Design

Retrospective, observational study.

### Setting

Tertiary-care, Level 1 trauma center.

### Patients

Adult patients with a PICC removed 1/1/2017 to 5/4/2020.

### Measurements

Frequency of PICC removal within five days of insertion and associated risk factors for early removal.

### Results

Between 1/1/2017 and 5/4/2020, 995 of 5348 PICCs inserted by the IV nursing team were removed within five days (19%). In 2017, 5 of 429 PICCs developed a central line-associated infection (1.2%) and 29 of 429 PICCs developed symptomatic venous thromboembolism (6.7%). Patients with PICCs whose primary service was in an ICU were independently at higher risk of early removal (OR 1.44, 95% CI 1.14, 1.83); weekday insertion was

**Data Availability Statement:** All relevant data are within the manuscript and its Supporting Information files.

**Funding:** The authors received no specific funding for this work.

**Competing interests:** I have read the journal's policy and the authors of this manuscript have the following competing interests: Dr. Mermel serves as a consultant for Light Line Medical Citius Pharmaceutical, and Destiny Pharma. This does not alter our adherence to PLOS ONE policies on sharing data and materials. The other authors have indicated they have no potential conflicts of interest to disclose.

independently associated with a lower likelihood of early removal compared to weekend insertion (OR 0.60; 95% CI 0.49, 0.75).

## Limitation

PICC removal after discharge was not assessed and paper records were likely incomplete and biased.

## Conclusion

Nearly one in five PICCs were removed within five days. Patients whose primary team was in an ICU and PICCs ordered on weekends were at independently higher risk of early removal.

## Introduction

Peripherally-inserted central catheters (PICCs) are an important component of medical care [1, 2]. They are frequently utilized when long-term intravenous (IV) access is required for antibiotic administration, medications requiring central venous access, parenteral nutrition, chemotherapy, or frequent blood draws in patients with difficult venous access [3]. Use of PICCs has been rising since 2001 due to their increased availability, ease of insertion, safety compared with central venous catheters (CVCs) inserted at other sites, and durability over extended periods of time [1, 4, 5]; however, the average dwell time for PICCs has been decreasing [5].

In hospitalized patients, insertion of a PICC for infusion of peripherally-compatible fluids, frequent phlebotomy, or difficult venous access is considered inappropriate if the expected duration of catheterization is five days or less [2]. However, infusion of irritants or vesicants such as parenteral nutrition or chemotherapy is appropriate for any proposed duration [2]. Two prospective, multi-center studies found that approximately 25% of PICCs in hospitalized patients had dwell times of five days or less [3, 6]. Known complications from PICCs include venous thromboembolism (VTE), central line-associated bloodstream infection (CLABSI), exit site infection, catheter lumen occlusion, and catheter tip migration, all of which are a cause of significant morbidity, potential mortality, and increased healthcare cost [7–11]. Placing PICCs that are removed within five days may be introducing unnecessary risk and preventable harm to a patient when there are safer alternatives [9, 10, 12, 13].

Mitigating inappropriate, short-term use of PICCs has the potential for harm reduction and cost-savings. The goals of this study were to assess: frequency and risk factors associated with PICC dwell time of five or fewer days; complications resulting from PICC use and determine if these complications were associated with the number of catheter lumens.

## Methods

### Study design

We performed a retrospective, observational study at Rhode Island Hospital, a tertiary-care, Level 1 trauma center licensed for 719 beds. Data were collected from two sources: paper records collected by our IV nursing team and the electronic health record (EHR). The IV nursing team is a nurse-led team that serves as a vascular access consult service throughout the hospital. The team evaluates patients for short-term peripheral venous catheters, midline catheters, and PICCs. All PICCs placed by the IV nursing team are ultrasound-guided. The IV nursing team kept paper records of PICC insertions and removals through 12/31/2017. On 1/1/2018,

the IV nursing team transitioned to full utilization of the EHR. Two reviewers (BHS, LM) used the paper records from 1/1/17 to 12/31/17 to identify any patient who had a PICC removed during their hospital stay. The correct patient and hospital encounter was then identified in the EHR. We identified patient demographics and characteristics including age, sex, race, and body mass index (BMI) at the time of PICC insertion. We identified the service team oversee-ing the patient's care at the time the PICC was ordered. Possible care teams included internal medicine (teaching service), hospital medicine (non-teaching service), general surgery, ortho-pedic surgery, neurosurgery, other medicine subspecialty, or other surgical subspecialty ser-vice. Podiatry, otolaryngology, plastic surgery, orthopedic surgery, neurosurgery, and dental surgery are all separate training programs from other surgical subspecialities. Among these, orthopedics and neurosurgery are the only ones that regularly admit patients to their service at our hospital; the other services admit to the medicine service for co-management. We hypoth-esized that the ordering specialty team and the day of the week would be predictors of early PICC removal.

We also identified characteristics of each PICC including number of lumens, indications for insertion and removal, order date, and time of insertion and removal. Lastly, we identified any PICC-associated complications. Symptomatic venous thromboembolism (VTE) of the extremity used for PICC placement and CLABSI were considered major complications. Symp-tomatic VTE was defined as any patient with a PICC who developed swelling, redness or pain that prompted imaging and which confirmed a thrombus. For CLABSI, we used the CDC National Healthcare Safety Network (CDC/NHSN) definition [14]. Minor complications included catheter occlusion, superficial thrombosis, mechanical complications such as kinking or coiling of the catheter, exit site infection, or catheter tip migration. All the information col-lected was recorded and stored in a REDCap database. Data from 1/1/2018 to 5/4/2020 was obtained exclusively from the EHR. Data collected during this time included patient demo-graphic information and objective PICC data including number of lumens, order date and time, insertion date and time, and removal date and time. We were unable to obtain the service team at the time of insertion, indications for PICC insertion, or complications related to the PICC since this free text data is difficult to obtain from an automated data pull. An EHR auto-mated data pull was also obtained for the January 2017-December 2017 time period to com-pare with and validate the manually collected data. The Lifespan Institutional Review Board approved this project.

## Inclusion and exclusion criteria

Inpatients at least 18 years of age who had a PICC inserted by our IV nursing team on or after 1/1/2017 and removed before 5/4/2020 were included in this study. Patients were excluded if their PICC was placed at an outside hospital or by other services (e.g., interventional radiol-ogy), as these PICCs were not included in the paper records kept by the IV nursing team. The vast majority of inpatient PICCs at our facility are placed by the IV team. Central lines placed by intensive care unit teams are placed in the internal jugular, subclavian, or femoral veins. Interventional radiology is consulted for PICCs only in circumstances where the IV team is unable to successfully insert the catheter and another type of central line would not suffice. Due to the small number and unique circumstances, PICCs placed by interventional radiology were excluded from this study. PICCs removed due to complications were not excluded.

## Statistical analysis

Data were imported into SAS version 9.4 (SAS Institute Inc., Cary, NC) for data management and hypothesis testing. The assessment of complications related to PICC use was based on

**Table 1. Characteristics of patients and PICCs.**

| Variable | 1/1/2017-12/31/2017 | 1/1/2017-5/4/2020 |
|---|---|---|
| N | 429 | 5348 |
| Age, mean (SD) | 57 (17) | 59 (17) |
| BMI, mean (SD) | 28 (10) | 30 (9) |
| Gender, n (%) | | |
| Female | 209 (49) | 2524 (47) |
| Male | 220 (51) | 2824 (53) |
| Race, n (%) | | |
| White | 324 (76) | 4176 (78) |
| Black | 37 (8.6) | 497 (9.3) |
| Other | 57 (13) | 655 (12) |
| Unknown | 11 (2.6) | 20 (0.4) |
| Have comorbidities, n (%) | N/A | 4292 (80) |
| PICC lumens, n (%) | | |
| 1 | 41 (9.6) | 1273 (24) |
| 2 | 333 (78) | 3621 (68) |
| 3 | 55 (13) | 454 (8.5) |
| Median dwell time, d (IQR) | 8.0 (3.9–18) | 13.0 (6.0–42) |
| Indication, n (%) | | |
| Antibiotics | 82 (19) | N/A |
| Chemotherapy | 14 (3.3) | N/A |
| Difficult venous access | 157 (37) | N/A |
| Long-term venous access | 6 (1.4) | N/A |
| Medications requiring central venous access | 10 (2.3) | N/A |
| Multiple incompatible fluids | 27 (6.3) | N/A |
| Parenteral nutrition | 24 (5.6) | N/A |
| Unknown | 31 (7.2) | N/A |
| Multiple | 78 (18) | N/A |
| Service Team, n (%) | | |
| General Surgery | 46 (11) | 582 (19) |
| Hospital Surgery (non-teaching) | 59 (14) | N/A |
| Intensive Care Unit | 47 (11) | 447 (15) |
| Internal Medicine | 51 (12) | N/A |
| Medical Subspecialty | 132 (31) | 783 (26) |
| Neurosurgery | 20 (4.7) | 217 (7.1) |
| Orthopaedic Surgery | 21 (4.9) | 303 (9.9) |
| Other | 53 (12) | 731 (24) |
| Day of the Week, n (%) | | |
| Sunday | 56 (13) | 655 (12) |
| Monday | 77 (18) | 868 (16) |
| Tuesday | 62 (15) | 794 (15) |
| Wednesday | 55 (13) | 751 (14) |
| Thursday | 60 (14) | 825 (15) |
| Friday | 62 (15) | 826 (16) |
| Saturday | 57 (13) | 629 (12) |
| Weekend day, n (%)* | 175 (41) | 4064 (76) |

N/A indicates that the data was not available.

*Weekend day defined as Friday, Saturday, and Sunday.

PICCs inserted on or after 1/1/17 and removed before 12/31/17 collected from paper records. The assessment of risk factors for dwell time of five days or fewer was based on EHR data of PICCs inserted on or after 1/1/17 and removed before 5/4/20. Descriptive statistics were obtained for the study sample characteristics. Mean and standard deviation were reported for the continuous variables while frequency and percentage were reported for the binary and categorical variables. The prevalence of PICCs placed that were removed within 5 days, as well as the prevalence of major and minor PICC complications, were reported. Generalized estimating equations (GEE) were used to examine the factors associated with PICCs removed within five days of insertion (GEE with a binomial distribution and logit link). The primary factors examined were service team, day of the week that PICC insertion was ordered, and indications for PICC insertion. Patient sex, age, body mass index, race, comorbidities, number of lumens, and number of complications were considered as possible model covariates. Only those possible model covariates with p<0.05 in univariable analyses were included in the multivariable model. GEEs were also used (1) to examine which PICC complications were associated with the number of lumens (GEE with a negative binomial distribution and logit link); and (2) to examine whether the rate of early removal changed over time (GEE with a binomial distribution and logit link). Classical sandwich estimators were used to protect against possible model misspecification. A p-value < 0.05 was used to determine statistical significance.

## Results

Approximately 11% of PICCs were placed in patients who were in an intensive care unit; over half of the catheters were double lumen PICCs (Table 1).

From 1/1/2017 through 12/31/2017, 141 PICCs were removed within five days of insertion. Among these 141 PICCs, 105 (74%) were removed due to reasons unrelated to complications. Complications were defined as DVT, mispositioning of the PICC, or complication arising from insertion, such as the PICC being too long or too short. From 1/1/17 through 5/4/20, 995 of 5348 PICCs were removed within five days (19%, Table 2).

Intensive care units were independently associated with a higher likelihood of early PICC removal (OR 1.44, 95% CI 1.14, 1.83), while weekday insertion (Monday through Thursday) was independently associated with a lower likelihood of early removal (OR 0.60; 95% CI 0.49, 0.75, Table 3). Complications were not associated with the number of PICC lumens (Table 4). Interestingly, chemotherapy or medications requiring central access, such as those often used

**Table 2. Outcomes of patients with PICCs.**

| Outcome | 1/1/2017-12/31/2017 | 1/1/2017-5/4/2020 |
|---|---|---|
| PICC removal within 5 days of insertion | 141/429 (33%) | 995/5348 (19%) |
| PICCs removed within 5 days not due to complication | 105/141 (74%) | N/A |
| Symptomatic venous thromboembolism | 29/429 (7%) | N/A |
| Central line-associated bloodstream infection | 5/429 (1%) | N/A |
| Catheter occlusion | 77/429 (18%) | N/A |
| Superficial venous thrombosis | 7/429 (2%) | N/A |
| Mechanical complication | 13/429 (3%) | N/A |
| Exit site infection | 0/429 (0%) | N/A |
| Catheter tip migration | 44/429 (10%) | N/A |
| Number without major or minor PICC complications | 303/429 (71%) | N/A |

N/A: Not applicable.

**Table 3. Multivariable model assessing associations with early PICC removal using the EHR data.**

| Variable | OR (95% CI) | p-value |
|---|---|---|
| Female | 0.87 (0.72–1.05) | 0.14 |
| Age | 1.01 (1.00–1.02) | 0.01 |
| BMI | 1.00 (0.99–1.02) | 0.44 |
| White | 0.94 (0.74–1.21) | 0.64 |
| Co-morbidity | 1.03 (0.81–1.32) | 0.80 |
| Team | | < .001 |
| General Surgery | *Reference* | |
| Hospital Medicine | N/A | |
| Intensive Care Unit | 2.10 (1.53–2.87) | |
| Internal Medicine | N/A | |
| Medical Subspecialty | 1.22 (0.91–1.65) | |
| Neurosurgery | 1.06 (0.68–1.63) | |
| Ortho Surgery | 0.69 (0.44–1.09) | |
| Other | 1.06 (0.78–1.44) | |
| PICC lumens | 0.96 (0.80–1.15) | 0.66 |
| Day of week | | 0.001 |
| Sunday | *Reference* | |
| Monday | 0.67 (0.48, 0.94) | |
| Tuesday | 0.59 (0.41, 0.83) | |
| Wednesday | 0.53 (0.36, 0.77) | |
| Thursday | 0.63 (0.44, 0.89) | |
| Friday | 0.55 (0.39, 0.79) | |
| Saturday | 0.93 (0.65, 1.33) | |

in intensive care units, was not a risk factor for early PICC removal. The percentage of early PICC removal by service team is displayed in Table 5.

## Discussion

Nearly one in five PICCs were removed within five days, similar to prior studies. (3)

Patients whose primary team was an intensive care team were at greater risk of having early PICC removal. This is not unexpected, as a PICC may be used for chemotherapy for fewer than five days, or for critical care of a patient whose condition improves within five days of insertion. Indication for PICC insertion was only available for the 1/1/2017 to 12/31/2017 data due to limitations in procuring this data from the EHR. Approximately one in three PICC removals were on intensive care units. Aside from BMI in the univariable analysis, comorbidities as a group were not associated with earlier PICC removal. BMI has previously been shown

**Table 4. Complications associated with number of PICC lumens based on 2017 manual chart review[*].**

| Complication | IRR | 95% CI | p-value |
|---|---|---|---|
| Symptomatic venous thromboembolism | 1.02 | (0.94, 1.11) | 0.66 |
| Central line-associated bloodstream Infection | 0.88 | (0.73, 1.08) | 0.24 |
| Catheter occlusion | 1.04 | (0.99, 1.08) | 0.14 |
| Catheter tip migration | 0.99 | (0.92, 1.08) | 0.90 |
| No. complications | 1.02 | (0.98, 1.05) | 0.35 |

[*]Based on univariable GEEs and lumens treated as a count (i.e., 1, 2, or 3 PICC lumens).

**Table 5. Early PICC removal by service team.**

| Service Team | Early PICC Removal | |
| --- | --- | --- |
| | **1/1/2017-12/31/2017** | **1/1/2017-5/4/2020 (EHR Data)** |
| General Surgery, % (95% CI) | 28 (17–43) | 15 (12–18)* |
| Hospital Medicine (non-teaching) | 44 (32–57) | N/A |
| Intensive Care Units | 21 (12–35) | 27 (23–31) |
| Internal Medicine | 33 (22–47) | N/A |
| Medical Subspecialty | 39 (31–47) | 18 (15–21)* |
| Neurosurgery | 25 (11–48) | 16 (11–21)* |
| Orthopedic Surgery | 24 (10–46) | 11 (7.8–15)*† |
| Other | 26 (16–40) | 16 (14–19) |

Note: Based on univariable GEE; Post hoc pairwise comparisons between service teams were conducted within the models via orthogonal contrasts; P-values are adjusted for multiple comparisons using the Holm test.

* $P<0.05$ for comparisons to Intensive Care Units.

† $P<0.05$ for comparisons to Medicine Subspecialty.

to be associated with PICC complications, such as CLABSI [15]. We found that dwell times may differ based on day of the week, or whether it is a weekday or weekend. When the order for PICC insertion occurred on a weekday, there was a significantly lower likelihood of early PICC removal. This may reflect PICC insertion during the weekend when there was limited phlebotomy services and nursing assistance and PICC removal the following weekdays when services for peripheral IV placement and phlebotomy are more available.

We could not confirm that the number of PICC lumens is associated with increased risk of complications such as CLABSI; however, our study may have been underpowered to assess for these outcome measures [16]. Due to the retrospective design and limitations of available data, no *a priori* power analysis was performed. Similarly, the number of attempts for PICC placement was not readily available for all insertions and is another limitation of our dataset.

Data for this study was shared with the hospital administration leading to a change in the name of the IV team to the Vascular Access Team and a change in hospital policy occurred such that practitioners were no longer able to order PICC for insertion by the team. Instead, providers were able to order a vascular access consult so the team could assess the patient based on information provided and medical record review to make the best decision regarding the most appropriate vascular access for the patient. We are tracking vascular access to assess the impact of these changes in hopes of reducing early PICC removal and improving patient outcomes.

Our study has a number of limitations. The paper records in 2017 were incomplete due to user omission, absent medical record numbers, and difficulty with handwriting. Additionally, there is likely selection bias regarding which PICC removals were documented. It is unclear what direction the selection bias may lean towards, as there are myriad reasons why documenting could have been variable on any given day. Since the paper records did not include all patients whose PICC was removed, complications may be over or underrepresented. We only assessed PICCs that were removed during hospitalization. Thus, we did not assess the many PICCs removed from patients after hospital discharge.

## Conclusion

In conclusion, weekend day orders for PICC insertion and intensive care unit teams were independent risk factors for removal of a PICC within five days. Additionally, Orthopaedic

surgery had significantly fewer early PICC removals than medicine subspecialties. Further study is needed to confirm these relationships, especially to determine how weekend staffing and coverage may or may not contribute to PICC orders leading to removal within five days. Our findings may be helpful for hospital administration to reduce inappropriate PICC use.

## Supporting information

**S1 Data. EHR data.**
(TXT)

**S2 Data. REDCap data.**
(TXT)

## Acknowledgments

We would like to thank the hospital Vascular Access Team for their help in the initial data collection.

## Author Contributions

**Conceptualization:** Burton H. Shen, Leonard A. Mermel.

**Data curation:** Burton H. Shen, Lindsey Mahoney.

**Formal analysis:** Janine Molino.

**Methodology:** Burton H. Shen, Janine Molino, Leonard A. Mermel.

**Resources:** Leonard A. Mermel.

**Supervision:** Leonard A. Mermel.

**Writing – original draft:** Burton H. Shen.

**Writing – review & editing:** Burton H. Shen, Lindsey Mahoney, Janine Molino, Leonard A. Mermel.

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
