## [Decision Letter · Decision Letter 0]

5 Apr 2022

PONE-D-22-03666Risk Factors for Early PICC Removal: A Retrospective Study of Adult Inpatients at an Academic Medical CenterPLOS ONE

Dear Dr. Mermel,

Thank you for submitting your manuscript to PLOS ONE. After careful consideration, we feel that it has merit but does not fully meet PLOS ONE’s publication criteria as it currently stands. Therefore, we invite you to submit a revised version of the manuscript that addresses the points raised during the review process.

We look forward to receiving your revised manuscript.

Kind regards,

Marc O. Siegel, MD

Academic Editor

PLOS ONE

Journal Requirements:

(I have read the journal's policy and the authors of this manuscript have the following competing interests: Dr. Mermel serves as a consultant for Light Line Medical Citius Pharmaceutical, and Destiny Pharma.  The other authors have indicated they have no potential conflicts of interest to disclose.)

We note that you received funding from a commercial source: (Light Line Medical Citius Pharmaceutical, and Destiny Pharma)

Reviewers' comments:

Reviewer's Responses to Questions

**Comments to the Author**

1. Is the manuscript technically sound, and do the data support the conclusions?

Reviewer #1: Yes

Reviewer #2: Partly

2. Has the statistical analysis been performed appropriately and rigorously? 

Reviewer #1: No

Reviewer #2: Yes

3. Have the authors made all data underlying the findings in their manuscript fully available?

Reviewer #1: No

Reviewer #2: Yes

4. Is the manuscript presented in an intelligible fashion and written in standard English?

Reviewer #1: Yes

Reviewer #2: Yes

5. Review Comments to the Author

Reviewer #1: Thank you for your submission to PLOS ONE for peer review.

Risk Factors for Early PICC Removal: A Retrospective Study of Adult Inpatients at an Academic Medical Center

This is a single centre, retrospective observational study from a large academic, trauma facility in the USA describing early removal of PICCs in a small patient sample.

There is no established power analysis reported despite its retrospective design. This makes findings difficult for generalization with other patient cohorts.

Overall, it is fairly well written, but lacks some clarity amongst some of its content.

There is no discussion regarding number of attempts to successfully place PICCs amongst the patient populations. This has been well established as a determining factor for PICC-related complications. I would consider adding this relevant data for analysis and discussion (if available), particularly if the practice was localized to just the IV Team. If data not available, this should be at least mentioned in the discussion or limitations section.

RESULTS

There is no serious contextual discussion of the results - only 4 tables with minimal sentence structure. I would consider at least providing some reporting of the results with a paragraph of text discussion, highlighting the more significant findings of the study.

TABLES

Table 1 - there is no description of any underlying co-morbidities - please provide at least discussion of influential co-morbidities that may have impacted complications or patient outcomes.

What does “Lumens, mean (SD)” mean? - what is the relevance? You have already described the subset of catheters with single and multiple lumens in the above rows. Is this different from this data? Appears a little confusing.

Change “Days in” to ‘Dwell Time’

Table 2.

Please add N/A to table legend.

Table 3.

Please complete missing data cells - Age, White (I presume you mean Race), and Comorbidity are missing reported data. If unavailable, consider removing from table and consolidating.

Team - is this describing the specialty area the patients were admitted under? What is the relevance of this data considering the inserter are a non-physician team? I do not see the benefits or relevance of this data or how it is impactful to the findings - especially when some of the data dismissing/not available.

What does the italicized REFERENCE mean in the table? Is this missing data? Please clarify.

Indications for PICC insertion are missing - how does this provide any clarity on the reason for device insertion and required therapies amongst these patient groups? This could be considered a large oversight. If no data is available, consider removing from table and discuss the lack of reported data in results or discussion section of manuscript.

Please label “P” in tables 3 & 4 as “P-Value”

CONCLUSION

Not stated

REFERENCES

11/13 (85%) >4 years old. Consider more recent scientific evidence to support your findings.

There appears to be an overuse of one authors published works 8/13 (62%), considering the scope of current literature on PICC-related outcomes that have published in the last few years focusing on specific device-related outcomes. While I acknowledge some f these works are systematic reviews, this is also more recent evidence available. I would consider the authors utilize a little more diversity across their choice of supportive clinical evidence.

Please provide DOI’s for all references wherever available, otherwise an internet link to the citation.

IMAGES

None provided.

I have several comments and questions.

Why were IR-inserted PICCs (or any other departments for that matter) excluded from this study? Surely if this large academic, trauma facility has significant numbers of inserted PICC devices, the differences between the IV team insertions and other inserter areas (e.g. IR, ICU, etc.) may possibly show differences in characteristics and outcomes between inserters, highlighting variances in patient cohorts, comorbidities and related complication rates across these clinical settings.

There is no description or model of the “IV Team” - is it an interdisciplinary team or a nurse-led team? Is it a “PICC team only”. Please consider briefly discussing the style or model of the team within the facility, as this may be influential in patient and device-related outcomes. There are a number of publications now available that describe vascular access teams function, scopes of practice and outcomes - while this may not be the authors priority, it provides an established foundation around the requirements of device insertion in the facility by the ‘team’).

There is a noticeable amount of missing data, highlighting a lack of established IV therapy/vascular access data collection processes related to PICC insertion. Specific data points are now required by most EHR’s to ensure adequate official reporting of pre- and post-procedural outcomes.

There is no reported data describing vessel characteristics (vessel choice/location, use of ultrasound guidance, measured vessel size and associated catheter to vessel ratio (CVR), or a description of insertion techniques (modified Seldinger, direct puncture, etc.), or the type of devices used (polyurethane, silicone, antimicrobial or antithrombogenic materials). These are all relative to the success or failure of device-related outcomes, whether for short, medium or long-term access. Considering the contemporary data that is frequently collected and presented in new publications, this would be considered a minimal requirement to establish the baseline variables and provide a widescreen view.

Reviewer #2: Manuscript Number: PONE-D-22-03666

Full Title: Risk Factors for Early PICC Removal: A Retrospective Study of Adult Inpatients at an

Academic Medical Center

Declaring competing interests: The reviewer declares that there are no competing interests.

Manuscript summary: The authors reviewed medical records of patients of the 5438 PICC during 40 mo. and revealed the significant risk factor for early PICC removal was medical subspeciality.

Scientific comments:

1. Background: It seems better to explain why early PICC removal is inappropriate.

2. Introduction: last paragraph

The goals of this study were to assess: frequency and risk factors associated with

catheter dwell time of five or fewer days; complications resulting from PICC use and determine if these complications were associated with the number of catheter lumens; and to assess changes in PICC dwell time during the study period.

- It would be better to bring focus on risk factors on early removal. Delete “to assess changes in PICC dwell….”. Furthermore, it was not mentioned in results.

3. Methods: Give information of PICCs, eg. French of PICC and insertion methods, by ECG guided or measurement of arm length.

4. Results Table 1. Please give the proportion of Intensive care unit vs. general ward.

5. Table 1. It seems no need to give average and SD for number of lumens.

6. Table 1. Indwelling days should be given as median and range (or interquartile). In this data, SD is much larger than the mean value which suggests that the data is more likely not a normal distribution.

7. Table 1. No. of Complications: give actual number and percentage, not mean and SD.

8. Table 2. Can the authors give number of PICC removal within 5 days and not caused by complications? That’s the primary endpoint noted at Objective..

9. Table 3. Please consider giving results of univariate analysis.

10. Table 4. Please, spell-out abbreviated words.

11. Replace gender with sex through-out manuscript, which is more bioscientific term.

12. Add conclusion at the end of Discussion, please.

Summary: This is an interesting topic and can be helpful for reduction of inappropriate use of PICC.

Although briefly mentioned in this study, lumens effect on CLABSI can be another independent topic, because generally multiple lumens have more risk of CLABSI but there is no randomized controlled study.

6. PLOS authors have the option to publish the peer review history of their article (what does this mean?). If published, this will include your full peer review and any attached files.

Reviewer #1: No

Reviewer #2: **Yes: **Dong Jae Shim

---

## [Author Response · Author response to Decision Letter 0]

17 May 2022

May 15, 2022

Marc O. Siegel, MD

Academic Editor

PLOS ONE

2150 Pennsylvania Ave NW 

Washington, DC, 20037

Dear Dr. Siegel,

Thank you for reviewing and considering our manuscript. We appreciate the questions and suggested changes brought forth by the reviewers. We are submitting a tracked changes version of our manuscript as well as a clean copy. 

Please see below for the reviewer comments and our response.

Sincerely,

Leonard A. Mermel, DO, ScM, AM (Hon), FSHEA, FIDSA, FACP 

Professor of Medicine, Warren Alpert Medical School of Brown University 

Medical Director, Dept. of Epidemiology & Infection Prevention, Lifespan Hospital System

Adjunct Clinical Professor, University of Rhode Island College of Pharmacy

There is no established power analysis reported despite its retrospective design. This makes findings difficult for generalization with other patient cohorts.

• As this was a retrospective study, we were limited by the data available. As such, no a priori power analysis was performed. We have added this limitation in the Discussion section. 

There is no discussion regarding number of attempts to successfully place PICCs amongst the patient populations. This has been well established as a determining factor for PICC-related complications. I would consider adding this relevant data for analysis and discussion (if available), particularly if the practice was localized to just the IV Team. If data not available, this should be at least mentioned in the discussion or limitations section.

• This data was not available for all PICC insertions. We have added this as a limitation in the Discussion section. 

RESULTS

There is no serious contextual discussion of the results - only 4 tables with minimal sentence structure. I would consider at least providing some reporting of the results with a paragraph of text discussion, highlighting the more significant findings of the study.

• We have added text highlighting the major findings of the study. 

TABLES

Table 1 - there is no description of any underlying co-morbidities - please provide at least discussion of influential co-morbidities that may have impacted complications or patient outcomes.

• We had added a line describing BMI as a significant co-morbidity during univariable analysis. 

What does “Lumens, mean (SD)” mean? - what is the relevance? You have already described the subset of catheters with single and multiple lumens in the above rows. Is this different from this data? Appears a little confusing.

• We agree that this part of the table was confusing and we have removed it. 

Change “Days in” to ‘Dwell Time’

• We have changed “days in” to “dwell time” throughout the manuscript. 

Table 2.

Please add N/A to table legend.

• We have added “N/A” to the table legend.

Table 3.

Please complete missing data cells - Age, White (I presume you mean Race), and Comorbidity are missing reported data. If unavailable, consider removing from table and consolidating.

• We understand how our original Table 3 could have presented some confusion. In that original analysis, we examined the factors associated with PICCs removed within five days of insertion. Table 3 presented the results from the multivariable analysis but did not include text necessary for interpreting the table correctly. In that original analysis, we hypothesized that service team and day of the week PICC insertion was ordered were associated early PICC removal. These variables were considered our key independent variables. Sex, age, body mass index, race, comorbidities, and number of lumens were considered as possible model covariates to be included in the multivariable model. These possible covariates were examined first in univariable models with early PICC removal (yes/no) as the outcome variable. Possible covariates were included in the multivariable model only if their univariable p-value was < 0.05, in order to preserve degrees of freedom and power. Thus, the empty rows in Table 3 corresponded to the possible covariates that did not meet our univariable criteria for inclusion in the multivariable model. The N/As in that table represented the study variables of interest that were not available in the EHR data sample.

In the revised manuscript, we refined our multivariable analyses presented in Table 3 in an effort to simplify the presentation and bring clarity to the data. In our updated multivariable analyses, we included all possible model covariates regardless of their univariable p-value. This was done, as their inclusion did not substantially alter the results and allows the reader to see that they were not significantly associated with early PICC removal.

Team - is this describing the specialty area the patients were admitted under? What is the relevance of this data considering the inserter are a non-physician team? I do not see the benefits or relevance of this data or how it is impactful to the findings - especially when some of the data is missing/not available.

• The Team describes the specialty patient care team that ordered insertion of a PICC. One of our hypotheses was that there would be a difference between the teams ordering PICC insertion and early removal. We have added the 1/1/2017 to 12/31/2017 data to Table 3 (see above for in depth discussion about changes to Table 3). 

What does the italicized REFERENCE mean in the table? Is this missing data? Please clarify.

• The italicized REFERENCE is the control group against which the other groups are compared for the stastical analysis. 

Indications for PICC insertion are missing - how does this provide any clarity on the reason for device insertion and required therapies amongst these patient groups? This could be considered a large oversight. If no data is available, consider removing from table and discuss the lack of reported data in results or discussion section of manuscript.

• We have added the PICC insertion indication from 1/1/2017 to 12/31/2017. Since the data in our EHR for indication is free form text, it was not available for the EHR-only data. We have added this limitation to the Discussion section.

Please label “P” in tables 3 & 4 as “P-Value”

• We have changed “P” to “P-Value.”

CONCLUSION

Not stated

• We have added a Conclusion. 

REFERENCES

11/13 (85%) >4 years old. Consider more recent scientific evidence to support your findings.

There appears to be an overuse of one author’s published works 8/13 (62%), considering the scope of current literature on PICC-related outcomes that have published in the last few years focusing on specific device-related outcomes. While I acknowledge some of these works are systematic reviews, this is also more recent evidence available. I would consider the authors utilize a little more diversity across their choice of supportive clinical evidence.

• We are not aware of any data suggesting the referenced articles older than four years are out of date or less relevant. We have no relationship to Dr. Chopra, nor personal reasons to include his publications in our manuscript. Dr. Chopra is a leading investigator in PICC research and we believe the cited work is important. 

Please provide DOI’s for all references wherever available, otherwise an internet link to the citation.

• We have added DOI’s to references wherever available. 

Why were IR-inserted PICCs (or any other departments for that matter) excluded from this study? Surely if this large academic, trauma facility has significant numbers of inserted PICC devices, the differences between the IV team insertions and other inserter areas (e.g. IR, ICU, etc.) may possibly show differences in characteristics and outcomes between inserters, highlighting variances in patient cohorts, comorbidities and related complication rates across these clinical settings.

• The vast majority of inpatient PICCs at our facility are placed by the IV team. Central lines placed by ICU teams are placed in the internal jugular, subclavian, or femoral veins. Interventional radiology is consulted for PICC insertion only in circumstances where the IV team is unable to place the PICC and another type of central line would not suffice. Due to the small number of such events and unique circumstances, PICCs placed by interventional radiology were excluded from this study. We have added this information to the revised Methods section. 

There is no description or model of the “IV Team” - is it an interdisciplinary team or a nurse-led team? Is it a “PICC team only?” Please consider briefly discussing the style or model of the team within the facility, as this may be influential in patient and device-related outcomes. There are a number of publications now available that describe vascular access teams function, scopes of practice and outcomes - while this may not be the authors priority, it provides an established foundation around the requirements of device insertion in the facility by the ‘team’).

• We have added a description of the IV team to the Study Design subsection. Evaluation of the vascular team function, scope of practice, and outcomes are beyond the scope of this study. 

There is a noticeable amount of missing data, highlighting a lack of established IV therapy/vascular access data collection processes related to PICC insertion. Specific data points are now required by most EHR’s to ensure adequate official reporting of pre- and post-procedural outcomes.

• We agree and one of the motivations for this study was to examine reasons for PICC insertion and to standardize the procedure and indication for inserting PICCs. Data from this study was shared with hospital administration to adjust the process by which evaluation for PICC insertion is ordered (see revised Discussion). 

There is no reported data describing vessel characteristics (vessel choice/location, use of ultrasound guidance, measured vessel size and associated catheter to vessel ratio (CVR), or a description of insertion techniques (modified Seldinger, direct puncture, etc.), or the type of devices used (polyurethane, silicone, antimicrobial or antithrombogenic materials). These are all relative to the success or failure of device-related outcomes, whether for short, medium or long-term access. Considering the contemporary data that is frequently collected and presented in new publications, this would be considered a minimal requirement to establish the baseline variables and provide a widescreen view.

• A limitation of our data set was that vessel choice/location, measured vessel size and associated catheter to vessel ratio, insertion technique were not readily available in our EHR. All PICC insertions performed by the IV team are ultrasound-guided. These factors are also beyond the scope of our manuscript as we sought to determine whether medical teams and days of the week (proxy for workflow variation) had influence of early PICC removal. 

Scientific comments:

1. Background: It seems better to explain why early PICC removal is inappropriate.

• We have discussed the risks of PICCs in general and added why early PICC removal may be inappropriate to the revised Introduction section. 

2. Introduction: last paragraph

The goals of this study were to assess: frequency and risk factors associated with

catheter dwell time of five or fewer days; complications resulting from PICC use and determine if these complications were associated with the number of catheter lumens; and to assess changes in PICC dwell time during the study period.

- It would be better to bring focus on risk factors on early removal. Delete “to assess changes in PICC dwell….”. Furthermore, it was not mentioned in results.

• We have removed that sentence. 

3. Methods: Give information of PICCs, eg. French of PICC and insertion methods, by ECG guided or measurement of arm length.

• A limitation of our data set was that insertion methods, PICC French size, etc. were not readily available. All PICC insertions are ultrasound-guided and we have added that to the Study Design subsection. 

4. Results Table 1. Please give the proportion of Intensive care unit vs. general ward.

• We have included number of intensive care unit PICC placements in revised Table 1. In addition, we have separated out intensive care unit teams and repeated the statistical analysis with this new subgroup (see above for further discussion on Table 3). 

5. Table 1. It seems no need to give average and SD for number of lumens.

• We have removed the average and SD. 

6. Table 1. Indwelling days should be given as median and range (or interquartile). In this data, SD is much larger than the mean value which suggests that the data is more likely not a normal distribution.

• We have changed the data to median and interquartile range.

7. Table 1. No. of Complications: give actual number and percentage, not mean and SD.

• We have removed the means and SDs and have left the number of total complications and percentages in Table 2.

8. Table 2. Can the authors give number of PICC removal within 5 days and not caused by complications? That’s the primary endpoint noted at Objective.

• We have added a line in the table to show the number of PICCs removed within 5 days that were not caused by complications. We have also added an explanation defining “complications.” 

9. Table 3. Please consider giving results of univariate analysis.

• The decision on whether a variable was included in the multivariable regression model was two pronged, and depended upon whether it was a hypothesized relationship and on its univariable model results. Service team, day of the week, and indications were included in the multivariable model regardless of significance because they were hypothesized to have a relationship with the outcome (i.e., we were testing whether that relationship was true or not). For other variables, they were only retained in the multivariable model if they were statistically significant in the univariable model. All variables in the descriptive statistics section were tested as possible model predictors in the univariable models. 

10. Table 4. Please, spell-out abbreviated words.

• We have spelled out abbreviated words in the revised text.

11. Replace gender with sex through-out manuscript, which is more bioscientific term.

• We have replaced “gender” with “sex” throughout the revised manuscript.

12. Add conclusion at the end of Discussion, please.

• We have added a Conclusion to the revised manuscript.

---

## [Decision Letter · Decision Letter 1]

12 Jun 2022

PONE-D-22-03666R1Risk Factors for Early PICC Removal: A Retrospective Study of Adult Inpatients at an Academic Medical CenterPLOS ONE

Dear Dr. Mermel,

Thank you for submitting your manuscript to PLOS ONE. After careful consideration, we feel that it has merit but does not fully meet PLOS ONE’s publication criteria as it currently stands. Therefore, we invite you to submit a revised version of the manuscript that addresses the points raised during the review process.

We look forward to receiving your revised manuscript.

Kind regards,

Marc O. Siegel, MD

Academic Editor

PLOS ONE

Journal Requirements:

Reviewers' comments:

Reviewer's Responses to Questions

**Comments to the Author**

1. If the authors have adequately addressed your comments raised in a previous round of review and you feel that this manuscript is now acceptable for publication, you may indicate that here to bypass the “Comments to the Author” section, enter your conflict of interest statement in the “Confidential to Editor” section, and submit your "Accept" recommendation.

Reviewer #1: All comments have been addressed

Reviewer #2: (No Response)

2. Is the manuscript technically sound, and do the data support the conclusions?

Reviewer #1: Yes

Reviewer #2: Partly

3. Has the statistical analysis been performed appropriately and rigorously? 

Reviewer #1: Yes

Reviewer #2: Yes

4. Have the authors made all data underlying the findings in their manuscript fully available?

Reviewer #1: Yes

Reviewer #2: Yes

5. Is the manuscript presented in an intelligible fashion and written in standard English?

Reviewer #1: Yes

Reviewer #2: Yes

6. Review Comments to the Author

Reviewer #1: Thank you for providing a revised version of your manuscript and addressing all the reviewers feedback.

This has strengthened the submission.

Reviewer #2: Manuscript Number: PONE-D-22-03666R1

Full Title: Risk Factors for Early PICC Removal: A Retrospective Study of Adult Inpatients at an

Academic Medical Center

Manuscript summary: The authors reviewed medical records of patients of the 5348 PICC during 40 mo. and revealed the significant risk factor for early PICC removal was patients whose primary team

was in an ICU and PICCs ordered on weekends. The manuscript revised well according to the comments, however, requires more revisions.

Scientific comments:

1. Abstract Results: “Patients with PICCs whose primary service was a medical subspecialty were independently at higher risk of early removal” This result should be updated according to new results.

2. Results: “In the 1/1/2017 to 12/31/2017 data set, female sex was associated with earlier PICC removal.” In this sentence, there were no p-values. Meanwhile, in table 3, female sex is not significantly associated with early removal (p=0.14). P-value of dataset of 2017 was not given. Please, clarify what lead to these results.

3. Results: “Approximately one-third of PICCs were placed in patients who were in an intensive care unit” In table 1, 30.8% of patients were medicine sub-speciality. ICU was 11%. There seems to be a confusion.

4. Results: “From 1/1/2017 through 12/31/2017, 141 PICCs were removed within five days of insertion. Among these 141 PICCs, 105 (74%) were not removed due to complications.” This sentence is inappropriate. “105 (74%) were removed from unrelated to complications” would be right.

5. Results: Table 1 shows total patient number as 7358. It should be clarified which analysis was applied for 7358 patients’ group. In results section, there seems to be no analysis on 7358 Pts. If there is no analysis on 7358, table 1 should include data of 5348 patients.

Grammar/style comments

6. Abstract Results:: Between 1/1/17 and 5/4/2020 -> Between 1/1/2017 and 5/4/2020

7. Discussion: Aside from BMI in the univariable analysis, comorbidities as a group were not associated with earlier PICC removal. BMI has previously been shown to be associated with PIC complications, such as CLABSI.[15] -> PICC

8. Table 3. Female 0.87 (0.7201.05) -> 0.87 (0.72-1.05)

Summary: This study had three dataset, first 429, second 5348, third 7358. This can cause reader’s confusion. It’d be better to simplify data set.

7. PLOS authors have the option to publish the peer review history of their article (what does this mean?). If published, this will include your full peer review and any attached files.

Reviewer #1: No

Reviewer #2: **Yes: **DONG JAE SHIM

---

## [Author Response · Author response to Decision Letter 1]

23 Jun 2022

June 23, 2022

Marc O. Siegel, MD

Academic Editor

PLOS ONE

2150 Pennsylvania Ave NW 

Washington, DC, 20037

Dear Dr. Siegel,

Thank you for reviewing and considering our manuscript. We appreciate the questions and suggested changes brought forth by the reviewers. We are submitting a tracked changes version of our manuscript and a clean copy. 

Please see below for the reviewer comments and our response.

Scientific comments:

1. Abstract Results: “Patients with PICCs whose primary service was a medical subspecialty were independently at higher risk of early removal” This result should be updated according to new results.

• We have updated the abstract results.

2. Results: “In the 1/1/2017 to 12/31/2017 data set, female sex was associated with earlier PICC removal.” In this sentence, there were no p-values. Meanwhile, in table 3, female sex is not significantly associated with early removal (p=0.14). P-value of dataset of 2017 was not given. Please, clarify what lead to these results.

• We have removed this line since this was a remnant of the previous analysis.

3. Results: “Approximately one-third of PICCs were placed in patients who were in an intensive care unit” In table 1, 30.8% of patients were medicine subspeciality. ICU was 11%. There seems to be a confusion.

• We have changed it to the correct number (11%) for ICU.

4. Results: “From 1/1/2017 through 12/31/2017, 141 PICCs were removed within five days of insertion. Among these 141 PICCs, 105 (74%) were not removed due to complications.” This sentence is inappropriate. “105 (74%) were removed from unrelated to complications” would be right.

• We have made this correction. 

5. Results: Table 1 shows total patient number as 7358. It should be clarified which analysis was applied for 7358 patients’ group. In results section, there seems to be no analysis on 7358 Pts. If there is no analysis on 7358, table 1 should include data of 5348 patients.

• We agree that the table and data is confusing with the 7358 total PICCs and the analysis done on the 5348 PICCs removed. We have simplified Table 1 to include just the 5348 patients with PICCs removed. 

Grammar/style comments

6. Abstract Results: Between 1/1/17 and 5/4/2020 -> Between 1/1/2017 and 5/4/2020

• We have made this correction.

7. Discussion: Aside from BMI in the univariable analysis, comorbidities as a group were not associated with earlier PICC removal. BMI has previously been shown to be associated with PIC complications, such as CLABSI.[15] -> PICC

• We have made this correction.

8. Table 3. Female 0.87 (0.7201.05) -> 0.87 (0.72-1.05)

• We have made this correction. 

Leonard A. Mermel, DO, ScM, AM (Hon), FSHEA, FIDSA, FACP 

Professor of Medicine, Warren Alpert Medical School of Brown University 

Medical Director, Department of Epidemiology & Infection Prevention, Lifespan Hospital System Adjunct Clinical Professor, University of Rhode Island College of Pharmacy

---

## [Editor Report · Decision Letter 2]

27 Jun 2022

Risk Factors for Early PICC Removal: A Retrospective Study of Adult Inpatients at an Academic Medical Center

PONE-D-22-03666R2

Dear Dr. Mermel,

We’re pleased to inform you that your manuscript has been judged scientifically suitable for publication and will be formally accepted for publication once it meets all outstanding technical requirements.

Kind regards,

Marc O. Siegel, MD

Academic Editor

PLOS ONE
---

## [Editor Report · Acceptance letter]

30 Jun 2022

PONE-D-22-03666R2 

Risk Factors for Early PICC Removal: A Retrospective Study of Adult Inpatients at an Academic Medical Center 

Dear Dr. Mermel:

I'm pleased to inform you that your manuscript has been deemed suitable for publication in PLOS ONE. Congratulations! Your manuscript is now with our production department. 

Kind regards, 

on behalf of

Dr. Marc O. Siegel 

Academic Editor

PLOS ONE